# Evolution of sex differences in cooperation can be explained by trade-offs with dispersal

**Pablo Capilla-Lasheras**[1,2,3]*, **Nina Bircher**[4], **Antony M. Brown**[1], **Xavier Harrison**[1], **Thomas Reed**[5], **Jennifer E. York**[6,7], **Dominic L. Cram**[6], **Christian Rutz**[8], **Lindsay Walker**[1], **Marc Naguib**[4], **Andrew J. Young**[1]*

**1** Centre for Ecology and Conservation, University of Exeter, Penryn, United Kingdom, **2** Swiss Ornithological Institute, Bird Migration Unit, Sempach, Switzerland, **3** School of Biodiversity, One Health and Veterinary Medicine, University of Glasgow, Glasgow, Scotland, **4** Behavioural Ecology Group, Wageningen University & Research, Wageningen, the Netherlands, **5** School of Biological, Earth and Environmental Sciences, University College Cork, Cork, Ireland, **6** Department of Zoology, Downing Street, University of Cambridge, Cambridge, United Kingdom, **7** Department of Zoology and Entomology, University of Pretoria, Pretoria, Republic of South Africa, **8** Centre for Biological Diversity, School of Biology, University of St Andrews, Sir Harold Mitchell Building, St Andrews, United Kingdom

* pacapilla@gmail.com (PC-L); A.J.Young@exeter.ac.uk (AJY)

**Data Availability Statement:** All R scripts and datasets needed to reproduce the analyses presented in this paper are available at: https://doi.org/10.5281/zenodo.13623047.

## Abstract

Explaining the evolution of sex differences in cooperation remains a major challenge. Comparative studies highlight that offspring of the more philopatric sex tend to be more cooperative within their family groups than those of the more dispersive sex but we do not understand why. The leading "Philopatry hypothesis" proposes that the more philopatric sex cooperates more because their higher likelihood of natal breeding increases the direct fitness benefits of natal cooperation. However, the "Dispersal trade-off hypothesis" proposes that the more dispersive sex cooperates less because preparations for dispersal, such as extra-territorial prospecting, trade-off against natal cooperation. Here, we test both hypotheses in cooperatively breeding white-browed sparrow weavers (*Plocepasser mahali*), using a novel high-resolution automated radio-tracking method. First, we show that males are the more dispersive sex (a rare reversal of the typical avian sex difference in dispersal) and that, consistent with the predictions of both hypotheses, females contribute substantially more than males to cooperative care while within the natal group. However, the Philopatry hypothesis cannot readily explain this female-biased cooperation, as females are not more likely than males to breed within their natal group. Instead, our radio-tracking findings support the Dispersal trade-off hypothesis: males conduct pre-dispersal extra-territorial prospecting forays at higher rates than females and prospecting appears to trade-off against natal cooperation. Our findings thus highlight that the evolution of sex differences in cooperation could be widely attributable to trade-offs between cooperation and dispersal; a potentially general explanation that does not demand that cooperation yields direct fitness benefits.

**Funding:** The long-term field study was funded principally by BBSRC David Phillips and NERC Blue Skies Research Fellowships to AJY (BB/H022716/1 and NE/E013481/1) and PC-L was supported by a BBSRC-funded PhD studentship (BB/M009122/1). A BBSRC David Phillips Research Fellowship to CR (BB/G023913/1 and BB/G023913/2) funded the Encounternet equipment. NB was supported by an ALW-NWO open competition grant (ALWOP 824.15.012) to MN. The funders had no role in study design, data collection and analysis, decision to publish, or preparation of the manuscript.

**Competing interests:** The authors have declared that no competing interests exist.

**Abbreviations:** GdLMM, generalised linear mixed model; LRS, lifetime reproductive success; LRT, likelihood-ratio test; RSSI, received signal strength indicator.

## Introduction

The evolution of sex differences in cooperation remains a major evolutionary puzzle [1–6]. In many cooperatively breeding species, in which offspring of both sexes delay dispersal from their natal group and help to rear future generations of their parents' young, one sex contributes significantly more than the other to cooperative care within the natal group ([1,3–5,7,8]; see also [3,4] for exceptions). These sex differences in natal cooperation are of particular interest as they occur in the absence of sex differences in relatedness to recipients (as helpers of both sexes are still residing within their natal groups), leaving it unlikely that kin selection alone can explain their evolution [1,3–6]. Recent comparative analyses have highlighted that, across cooperative breeders of this kind, sex differences in contributions to cooperative care while within the natal group are predicted by the species' sex difference in dispersal: resident offspring of the more philopatric sex tend to contribute more to natal cooperation than resident offspring of the more dispersive sex [3,4]. However, it is not clear why the more philopatric sex contributes more to natal cooperation, as the two major evolutionary explanations proposed for this pattern have not been teased apart [1,3–5,9].

The leading hypothesis for the evolution of sex differences in natal cooperation, the "Philopatry hypothesis," proposes that offspring of the more philopatric sex help at higher rates (while both sexes reside within the natal group) because, by staying in their natal group for longer they stand to gain a greater direct fitness benefit from cooperation [1,3,6]. This hypothesis therefore requires that cooperation yields a direct fitness benefit and that the accrual of this benefit is to some extent contingent upon remaining within the natal group. For example, the more philopatric sex may be more likely to ultimately breed within the natal territory and may thus stand to gain a greater downstream direct benefit from cooperating to rear additional group members that may ultimately help them in the future [1–3,6]. A recent comparative study of the cooperative birds in which both sexes delay dispersal and help has found support for this idea [3]; species with more female-biased probabilities of natal breeding showed more female-biased helper contributions to cooperative care while within the natal group. These findings are consistent with the key prediction of the Philopatry hypothesis and could therefore constitute rare evidence that direct fitness benefits have played a widespread role in the evolution of helping [3,10]. However, these findings are also consistent with an alternative less-studied hypothesis which also predicts that the more philopatric sex should help more, regardless of whether helping yields a direct benefit, the "Dispersal trade-off hypothesis."

The Dispersal trade-off hypothesis proposes that offspring of the more dispersive sex help at lower rates (while both sexes reside within the natal group) because all individuals face a trade-off between investments in natal cooperation and activities that promote dispersal from the natal group, such as extra-territorial prospecting for dispersal opportunities [5,9]. Thus, while the Philopatry hypothesis requires that cooperation yields a downstream direct fitness benefit, the Dispersal trade-off hypothesis does not. In many social species, individuals prepare for dispersal by conducting extra-territorial prospecting forays from their natal group, which are thought to yield information on dispersal opportunities in the surrounding population [5,6,9,11–14]. Investments in prospecting could trade-off against natal cooperation because the two activities cannot be carried out at the same time and because costs and physiological changes associated with prospecting could reduce the expression of helping [9,12,14–16]. Frequent prospectors could also reduce their costly contributions to helping in order to maintain competitive phenotypes that reduce the risks and/or increase the likely success of prospecting [17]. However, little is known about prospecting due to the difficulty of monitoring the cryptic, fast, and often long-distance movements that prospectors make [5,9,11,13,15]. One study of male meerkats, *Suricata suricatta*, whose frequent extra-territorial forays yield extra-group

matings [18], suggests that investments in such forays may indeed trade-off against cooperation [9]. But whether prospecting for dispersal opportunities per se (which may occur at lower rates than mating forays and could entail different costs) can also trade-off against natal cooperation, as envisaged under the Dispersal trade-off hypothesis, is not yet known.

As the Philopatry and Dispersal trade-off hypotheses for the evolution of sex differences in cooperation both predict that helpers of the more philopatric sex should contribute more to cooperative care while within the natal group (consistent with the findings of comparative studies [3,4]), discriminating between them now requires innovative field studies to test whether the different underlying processes assumed by each hypothesis are acting in the manner that would be required to explain the observed sex differences in cooperation.

Here, we combine long-term data on life-histories and cooperative behaviour with a high-resolution automated radio-tracking study of prospecting to test both the predictions and core assumptions of the Philopatry and Dispersal trade-off hypotheses, using wild cooperatively breeding white-browed sparrow weavers, *Plocepasser mahali*, as a model system. White-browed sparrow weavers (hereafter "sparrow weavers") are cooperative birds that live in social groups and defend year-round territories across the semi-arid regions of sub-Saharan Africa. Within each group, a single dominant male and female monopolise reproduction and their offspring of both sexes delay dispersal, becoming nonbreeding subordinates (occurring at an approximately even sex ratio; [19]) that cooperatively help to feed future generations of nestlings produced by the breeding pair [20]. As offspring of both sexes never breed while subordinate [20], the only route to natal breeding is via inheritance of the natal dominant (breeding) position. The primary route to dominance for both sexes, however, is via dispersal to other groups [21], which may offer immediate access to vacant dominant positions and is likely to reduce the risk of inbreeding [20,22,23]. Dispersal is typically local in both sexes but genetic analyses and field observations suggest that males disperse further than females from their natal to breeding sites and thus that females may be the more philopatric sex [21], a rare reversal of the typical passerine sex-bias in dispersal [24]. Here, we establish whether there are also clear sex biases in dispersal incidence and natal breeding position inheritance, the dispersal traits most relevant to the rationale of the focal hypotheses. Subordinates of both sexes conduct extra-territorial prospecting forays prior to dispersal from their natal group, which are commonly met with aggression from territory holders [21,25]. The rarity of extra-group parentage by subordinates (<1% of paternity [20]) suggests that the primary function of these forays is to promote dispersal.

Subordinate female sparrow-weavers cooperatively feed offspring at substantially higher rates than subordinate males [19], a pattern rarely documented among birds [5]. Here, we demonstrate that this marked sex difference in cooperation is evident among subordinates within their natal groups, in the absence of a sex difference in relatedness to recipients (see Results for both findings). Helper contributions are known to increase the overall rates at which nestlings are fed [19]. This appears to increase offspring survival to fledging in dry conditions but actually reduce it in wet conditions [19], such that helping reduces the rainfall-induced variance in offspring survival to fledging (which may yield indirect fitness benefits via mechanisms such as altruistic bet-hedging; [19]) without impacting the mean level of offspring survival to fledging once one integrates across environmental conditions [19]. Helping may also yield indirect fitness benefits by lightening the workloads of related breeders [26,27], thereby improving the survivorship of the dominant female [28] and allowing her to invest more per offspring at the egg stage when helped [29]. The combined effects of helping on maternal investment at the egg stage and the total rate of nestling provisioning could also yield indirect fitness benefits by improving offspring quality [26] but this remains to be tested. Helping could also conceivably yield direct fitness benefits, but as subordinate immigrants rarely

help at all [27], it seems unlikely that these benefits are substantial (see Discussion for more detail). Helping behaviour could, for example, yield a downstream direct fitness benefit to helpers if they inherited the dominant (breeding) position in their natal group and were then helped by offspring that they had previously helped to rear; the mechanism commonly envisaged in the Philopatry hypothesis [3,6].

First, we demonstrate that the sex differences in dispersal and cooperation in sparrow-weaver societies satisfy the predictions of both the Philopatry and Dispersal trade-off hypotheses: This species shows a reversal of the typical avian sex biases in both dispersal incidence and natal cooperation. We then test for the underlying processes assumed by the Philopatry and Dispersal trade-off hypotheses to establish whether either hypothesis can explain why this rare example of female-biased Philopatry in birds is accompanied by female-biased natal cooperation. Regarding the Philopatry hypothesis, we test whether female helpers are indeed more likely than males to inherit the breeding position within their natal group; the commonly invoked mechanism by which the more philopatric sex could gain a greater downstream direct benefit from cooperation [1,3]. Regarding the Dispersal trade-off hypothesis, we use a high-resolution automated radio-tracking study of extra-territorial prospecting to test its two key assumptions: first, that male helpers (the more dispersive sex) prospect at higher rates or over greater distances or for longer durations than females; second, that investments in prospecting from the natal group trade-off against concurrent investments in natal cooperative care.

## Results

### Testing the predictions of the Philopatry and Dispersal trade-off hypotheses

Consistent with the predictions of both hypotheses, sparrow weavers show a significant male-bias in dispersal incidence coupled with a significant female-bias in contributions to natal cooperation. Modelling the age-specific probability of dispersal from the natal group revealed that subordinate natal males are significantly more likely to disperse than subordinate natal females across all age classes ($\chi^2_1$ = 9.64, $p$ = 0.002; Fig 1A and Table 1). There was also evidence that dispersal probability increased with age ($\chi^2_3$ = 64.98, $p$ < 0.001; Fig 1A and Table 1), but no evidence that the sex difference in dispersal probability changed with age ($\chi^2_3$ = 1.35, $p$ = 0.717; Table 1). The Philopatry and Dispersal trade-off hypotheses would therefore both predict that subordinate females (the more philopatric sex) should help at higher rates than subordinate males (the more dispersive sex) while within the natal group.

Subordinate females did indeed feed broods at significantly higher rates than subordinate males while within the natal group ($\chi^2_1$ = 16.69, $p$ < 0.001; Fig 1B and Table 2). The provisioning rates of subordinates also increased initially with advancing age before decreasing slightly (Fig 1B; age effect: $\chi^2_3$ = 27.41, $p$ < 0.001; Table 2). There was no indication that the magnitude of the sex difference in cooperation changed significantly with age ($\chi^2_3$ = 5.41, $p$ = 0.162; Table 2). Brood size and brood age both positively predicted subordinate provisioning rates (both $\chi^2_1$ > 11.50, $p$ < 0.001; Table 2). During their provisioning visits, subordinate females also spent slightly but significantly longer than subordinate males within the nest chamber ($\chi^2_1$ = 16.46, $p$ < 0.001; Fig 1C and S1 Table) and were just as likely as subordinate males to provision nestlings with food items that were large ($\chi^2_1$ = 1.43, $p$ = 0.231; Fig 1D and S2 Table). These sex differences in the provisioning rates of subordinates within the natal group occurred in the absence of sex differences in their relatedness to the nestlings being fed ($\chi^2_1$ = 0.16, $p$ = 0.691; Fig 1E and S3 Table).

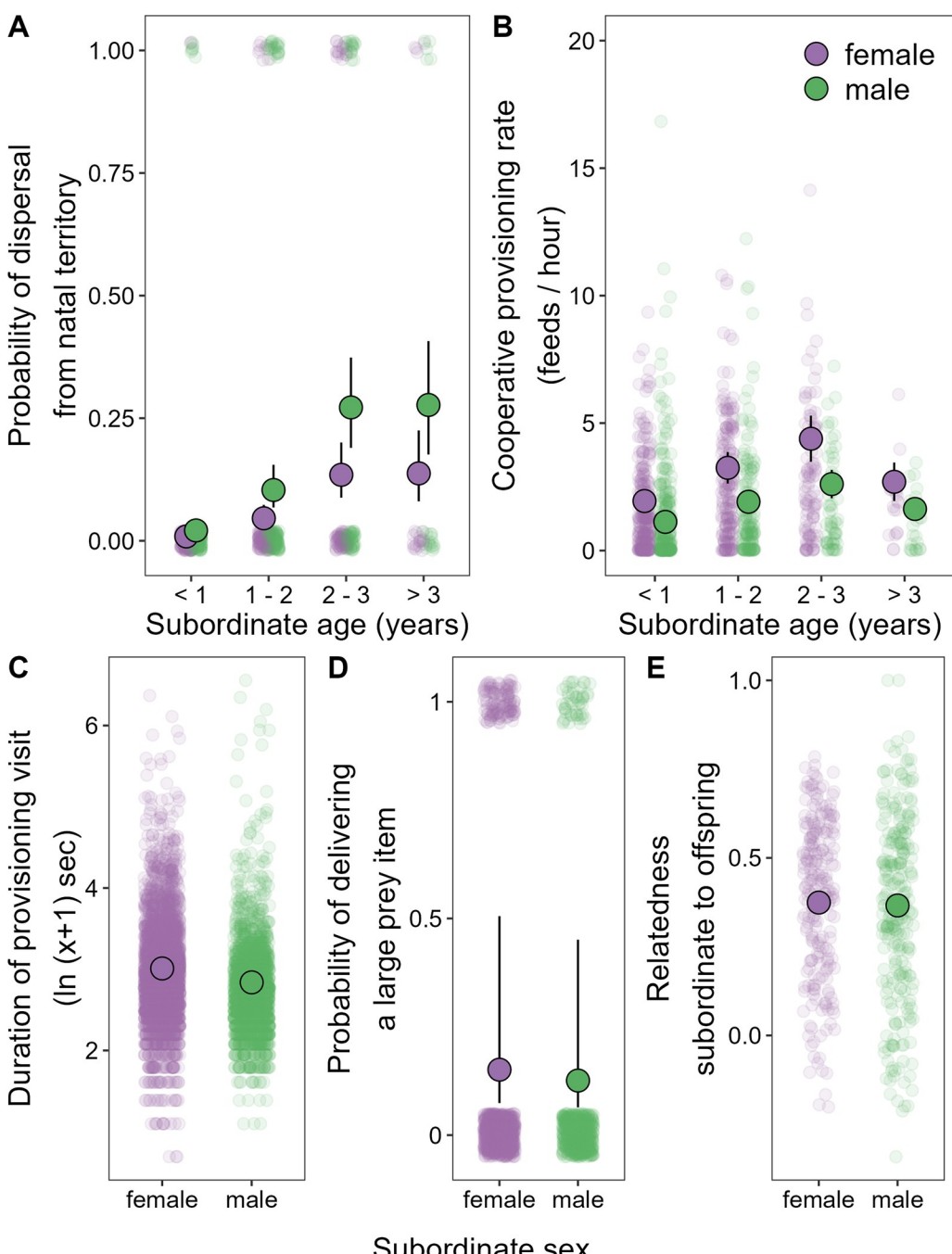

**Fig 1. Sparrow weavers show a rare reversal of the typical avian sex differences in dispersal and natal cooperation.** Among subordinate birds residing within their natal groups, (**A**) males (green points) showed significantly higher age-specific probabilities of dispersal than females (purple points; Table 1), while (**B**) females showed significantly higher cooperative nestling provisioning rates than males (a sex difference that did not vary with age; Table 2). (**C**) Natal subordinate females also spent slightly but significantly longer in the nest per provisioning visit than males (S1 Table) and (**D**) were just as likely as natal subordinate males to feed the nestlings with food items that were large (S2 Table). (**E**) These sex differences in natal cooperation occurred in the absence of a sex difference in relatedness to the nestlings being fed (S3 Table). Translucent points represent raw data whereas larger points and error bars give mean model predictions ± SE (where error bars cannot be seen, they are smaller than the radius of the point presenting the mean). Data and code needed to generate this figure can be found at https://doi.org/10.5281/zenodo.13623047.

**Table 1. Coefficients and likelihood-ratio tests of binomial mixed model explaining variation in probability of age-specific dispersal from natal group ($n$ = 813 observations of 338 subordinates, 178 males and 160 females, 82 dispersal events).** The interaction between subordinate age and subordinate sex did not receive statistical support ($\chi^2_3$ = 1.35, $p$ = 0.717) and was dropped from the full model to ease interpretation of single effect predictors. Model coefficients are shown in the link-function scale ("logit").

| Fixed effect | Estimate | SE$^A$ | 95% CI$^A$ | $\chi^2$ | df$^A$ | $p$ |
|---|---|---|---|---|---|---|
| Intercept | −4.716 | 0.584 | −5.861, −3.570 | | | |
| **Subordinate sex** | | | | 9.64 | 1 | 0.002 |
| *Female* | — | — | — | | | |
| *Male* | 0.877 | 0.281 | 0.326, 1.427 | | | |
| **Subordinate age (years)** | | | | 64.98 | 3 | <0.001 |
| *< 1* | — | — | — | | | |
| *1–2* | 1.681 | 0.394 | 0.909, 2.453 | | | |
| *2–3* | 2.854 | 0.419 | 2.032, 3.676 | | | |
| *>4* | 2.879 | 0.563 | 1.775, 3.983 | | | |
| **Random effect variance** | **Estimate** | **# Levels** | | | | |
| Individual ID | 0.000 | 338 | | | | |
| Natal social group ID | 0.475 | 36 | | | | |
| Breeding season of hatching | 0.879 | 8 | | | | |

$^A$ SE = Standard Error, CI = Confidence Interval, df = degrees of freedom likelihood-ratio test.

## Testing the mechanisms invoked by the Philopatry hypothesis

Contrary to expectations under the Philopatry hypothesis, females were no more likely than males to obtain a dominant breeding position within their natal group ($\chi^2_1$ = 0.29, $p$ = 0.592; Fig 2 and S4 Table). For natal subordinates under 3 years of age, the probability of ultimately inheriting natal dominance was low (<10% on average, Fig 2) and showed no sex difference, yet sex differences in natal cooperation were clear at these ages (Fig 1B). Once subordinates reached 3 years of age within their natal group their probability of inheriting natal dominance markedly increased (to approximately 25%), but their contributions to natal cooperation markedly decreased (Fig 1B). If we also included in these "natal dominance acquisition" events any dominant breeding positions acquired outside the natal group by founding a new group within a territory previously held by the natal group (i.e., territorial budding [6]), we still found no evidence of a sex difference in the probability of natal dominant acquisition ($\chi^2_1$ = 1.89, $p$ = 0.169; S5 Table). Overall, dominance turnover events rarely involved subordinates winning dominance within their natal group, and the sexes did not differ in this regard (just 7 natal dominance wins of 32 total turnovers for males; 9 natal of 32 total for females; $\chi^2_1$ = 0.08, $p$ = 0.773); dominance was typically acquired in both sexes via dispersal to other groups. The observed male-bias in age-specific dispersal probability (Fig 1A) appears, therefore, to reflect a sex difference in the strategy used to acquire dominance via dispersal, rather than in the incidence of acquiring dominance via dispersal (see Text D in S1 File).

## Testing the mechanisms invoked by the Dispersal trade-off hypothesis

**Direct observations of prospecting.** On 184 occasions, resident subordinate birds in one group were observed visiting the territory of another group, where they were typically aggressively chased by the residents. Of these 184 observed prospecting forays, 115 (62.5%) were conducted by 68 males and 69 (37.5%) were conducted by 51 females, which suggests a male-bias in prospecting incidence given the approximately equal subordinate sex ratio in our population ([19]; 115 of 184 forays is a significant deviation from 50%; binomial test $p$ < 0.001).

**Table 2. Coefficients and likelihood-ratio tests of model explaining variation in the cooperative contributions to offspring provisioning by subordinates within their natal groups ($n$ = 1,338 cooperative provisioning rate estimates for 314 subordinates, 156 males and 159 females).** The response term was the number of provisioning visits by an individual during a given observation period, and the model included the duration of these observation periods as an offset (thereby effectively modelling provisioning rate; feeds/hour). The interaction between subordinate age and subordinate sex did not receive statistical support ($\chi^2_3$ = 5.14, $p$ = 0.162) and was dropped from the full model to ease interpretation of single effect predictors. The model accounted for zero-inflation (zero-inflation parameter = −1.605, SE = 0.181) and over-dispersion (negative binomial dispersion parameter = 2.87). Model coefficients are shown in the link-function scale ("log").

| Fixed effect | Estimate | SE[A] | 95% CI[A] | $\chi^2$ | df[A] | $p$ |
|---|---|---|---|---|---|---|
| Intercept | −1.164 | 0.371 | −1.891, −0.436 | | | |
| **Subordinate sex** | | | | 16.69 | 1 | <0.001 |
| *Female* | — | — | — | | | |
| *Male* | −0.522 | 0.128 | −0.773, −0.271 | | | |
| **Subordinate age (years)** | | | | 27.41 | 3 | <0.001 |
| *<1* | — | — | — | | | |
| *1–2* | 0.512 | 0.126 | 0.265, 0.760 | | | |
| *2–3* | 0.740 | 0.154 | 0.439, 1.041 | | | |
| *>4* | 0.317 | 0.247 | −0.167, 0.800 | | | |
| **Brood age (days)** | | | | 26.16 | 6 | <0.001 |
| *6* | — | — | — | | | |
| *7* | 0.201 | 0.263 | −0.315, 0.717 | | | |
| *8* | 0.364 | 0.289 | −0.202, 0.929 | | | |
| *9* | 0.308 | 0.284 | −0.248, 0.864 | | | |
| *10* | 0.574 | 0.276 | 0.033, 1.114 | | | |
| *11* | 0.584 | 0.277 | 0.040, 1.127 | | | |
| *12* | 0.803 | 0.277 | 0.259, 1.347 | | | |
| **Brood size** | 0.423 | 0.123 | 0.182, 0.664 | 11.58 | 1 | <0.001 |
| **Random effect variance** | Estimate | # Levels | | | | |
| Individual ID | 0.646 | 314 | | | | |
| Social group ID | 0.000 | 38 | | | | |
| Breeding season | 0.007 | 10 | | | | |
| Clutch ID | 0.393 | 177 | | | | |

[A] SE = Standard Error, CI = Confidence Interval, df = degrees of freedom likelihood-ratio test.

Given the potential for male and female prospectors to differ in how readily their forays can be detected by observers (e.g., due to sex differences in behaviour while prospecting), we then sought to verify this pattern with a radio-tracking study of subordinates.

**Automated tracking of prospecting behaviour.** We deployed an automated radio-tracking technology to provide continuous high-resolution information on the prospecting movements of 27 radio-tagged subordinate adults residing within their natal groups (13 males and 14 females, Fig 3A). After processing the 22,518,022 detection logs from the base station receiver array (see Methods), we detected a total of 971 extra-territorial prospecting forays. We used a conservative approach to identify forays (see Methods): events in which the signal-strength distribution across the receiver array suggested that the bird's closest base-station was >250 m away from the centre of their own territory (where there was also always a base-station). As base-stations were situated in the centres of all territories within the tracking study area and the mean ± SE distance between neighbouring territory centres was just 93.7 m ± 4.56 m, these events should typically constitute forays beyond the territory centres of neighbouring groups (see Methods). This approach will minimise the chance that a resident bird's territorial interactions with its neighbours are incorrectly interpreted as extra-territorial prospecting, but may underestimate foray frequency by excluding more local prospecting.

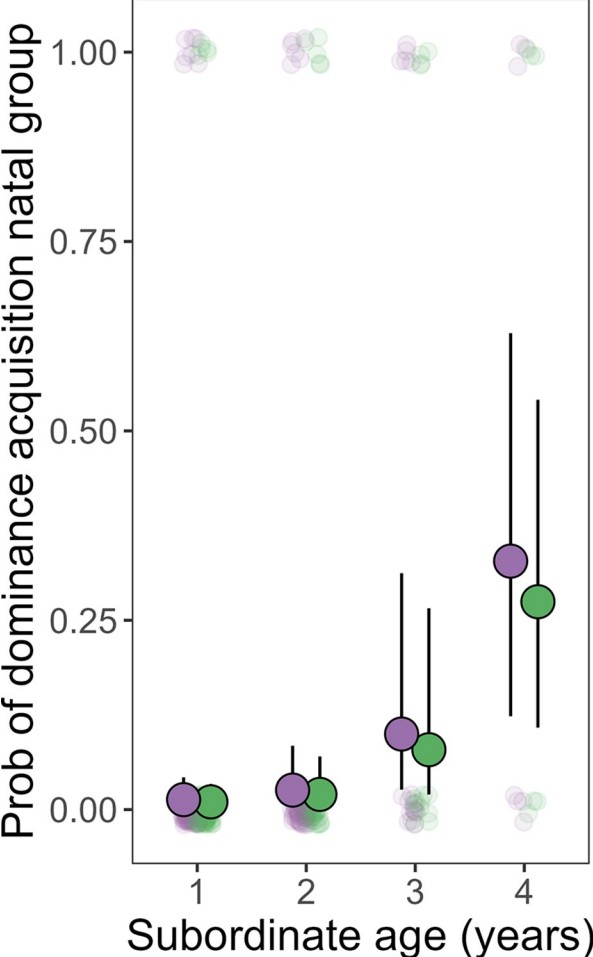

**Fig 2. Age-specific probabilities of inheriting a dominant breeding position within the natal group, for subordinates still residing within their natal group at various ages.** Subordinate females (purple points; the more cooperative sex; Fig 1B) were no more likely to inherit a natal breeding position than subordinate males (green points; S4 Table). Our analyses focussed on subordinates within their natal group at a given focal age and tested for an effect of sex on their probability of inheriting natal dominance at some later time. The solid points show the sex-specific means (± SE) for the focal age class, and the translucent points represent raw data. Subordinates were only included in the analysis if dominance was monitored within their focal group for at least 2 years after the date on which the subordinate reached the focal age. Data and code needed to generate this figure can be found at https://doi.org/10. 5281/zenodo.13623047.

These 971 forays principally occurred between 06:00 h and 19:00 h, with frequencies peaking in the cooler periods towards the start and end of the day (Fig 3B). The tagged bird's estimated daily rates of prospecting varied widely from 0 to 26 forays per day ($n$ = 895 daily rate measures from 27 tagged birds; mean = 1.09 forays/day; median = 1 foray/day). Forays had a mean duration of 18.59 min (median = 5.67 min; range = 0.42 to 493 min). Our best estimate of the round-trip distance travelled per foray had a mean of 720.74 m (median = 647.60 m; range = 502.29–2026.15 m), though this is likely an underestimate as our ability to detect longer-distance forays was constrained by the spatial coverage of our receiver array (S1 Fig), and as the estimate was calculated as twice the straight line distance to the bird's furthest location (while the birds could have taken a more complex path).

Consistent with our direct observations above, subordinate males showed significantly higher daily rates of prospecting from their natal groups than subordinate females ($\chi^2_1$ = 5.39,

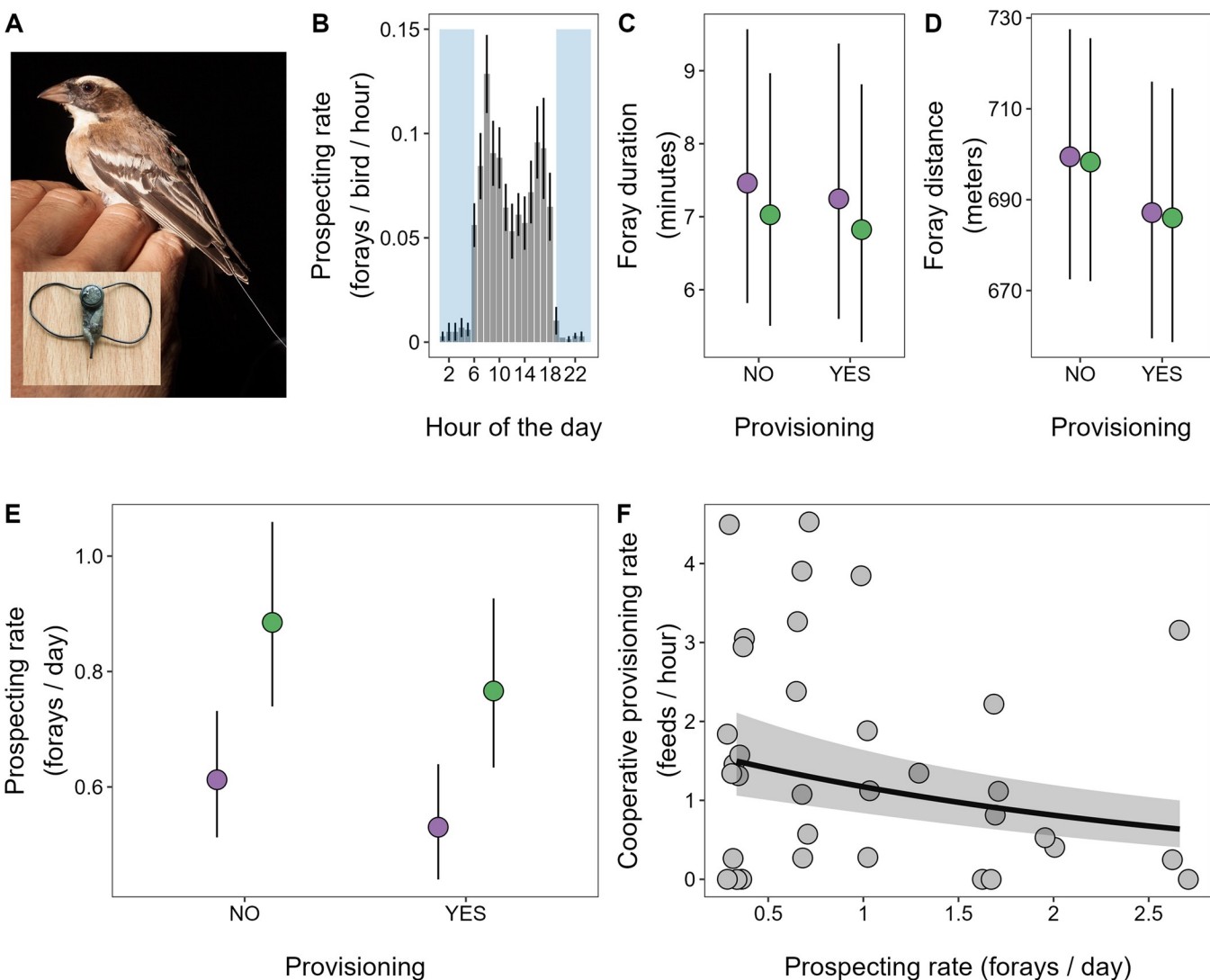

**Fig 3. Extra-territorial prospecting behaviour and its trade-off with cooperation.** (**A**) A subordinate female (females have pale bills) fitted with an Encounternet radio-tag (pictured inset). The tag is fitted with leg loops and sits on the bird's lower back. The aerial can be seen sitting along the tail feathers. (**B**) Circadian variation in prospecting behaviour; mean hourly prospecting rates per bird per hour of the day (mean ± SE; from the 27 radio-tracked birds). The shaded area illustrates night hours (using the local civil twilight time on March 11, 2017, the middle date of our tracking study). The rare forays at night could reflect birds being flushed from their roost chambers by predators. We found no evidence of sex differences in either the (**C**) mean duration of forays (S7 Table) or (**D**) mean estimated round-trip distance travelled on forays (S8 Table). (**E**) Subordinate males have a significantly higher daily prospecting rate than subordinate females, both inside and outside peak periods of cooperative nestling provisioning (S6 Table). (**F**) Individual variation in prospecting rate negatively predicted cooperative provisioning rate for the 13 radio-tagged birds with prospecting data during a nestling provisioning period (S9 Table; the plot shows >13 data points as we have repeated measures per individual). Male and female model estimates in **C**, **D**, and **E** illustrated in green and purple colour, respectively. **C**, **D**, **E**, and **F** present mean model predictions ± SE. Data and code needed to generate this figure can be found at https://doi.org/10.5281/zenodo.13623047.

$p = 0.020$; Fig 3E and S6 Table). Prospecting rates were modestly but not significantly lower during nestling provisioning periods (i.e., when cooperative care occurs) than at other times ($\chi^2_1 = 2.15$, $p = 0.142$; Fig 3E and S6 Table). The magnitude of the sex difference in prospecting rate was not affected by whether the bird's social group was provisioning nestlings at the time ($\chi^2_1 = 0.97$, $p = 0.324$; S6 Table legend). There was no evidence of an effect of subordinate sex or nestling provisioning periods on either the duration (both $\chi^2_1 < 0.09$, $p > 0.77$; Fig 3C and S7 Table) or estimated distance of forays (both $\chi^2_1 < 0.80$, $p > 0.37$; Fig 3D and S8 Table).

Older subordinates conducted forays of shorter duration ($\chi^2_1$ = 3.95, $p$ = 0.047; S7 Table) but age did not predict prospecting rate ($\chi^2_1$ = 0.23, $p$ = 0.631; S6 Table) or distance ($\chi^2_1$ = 0.14, $p$ = 0.708; S8 Table). Focussing on the subset of radio-tagged subordinates whose forays were tracked through a nestling provisioning period, we also found evidence suggestive of a trade-off between prospecting and cooperative provisioning. Despite the modest sample size available for this analysis (34 daily measures of provisioning rate with matched prospecting effort data, from 13 subordinates across 7 social groups), an individual's prospecting rate over the three-day window prior to us quantifying cooperative contributions to nestling provisioning significantly negatively predicted its cooperative provisioning rate ($\chi^2_1$ = 5.08, $p$ = 0.024; Fig 3F and S9 Table). The existence of this negative correlation between prospecting and cooperative provisioning rate was also robust to the choice of time window over which we calculated prospecting effort (S2 Fig).

## Discussion

Our findings reveal that white-browed sparrow weavers show a rare reversal of the typical avian sex difference in dispersal: males show higher age-specific probabilities of dispersal from their natal group than females and both population genetics and observed dispersals suggest that males disperse further than females from their natal to breeding sites [21]. Undetected longer-distance dispersals (beyond the bounds of our study site) may also be more common among males, as the low rate of immigration into our study population is also male biased [21]. Both the Philopatry and Dispersal trade-off hypotheses for the evolution of sex differences in cooperation would therefore predict that sparrow-weavers show female-biased natal cooperation, and our findings confirm this. Natal subordinate females helped to feed nestlings at higher rates than natal subordinate males across the age classes, and spent longer within the nest on their visits. This coupling of reversals of the typical avian sex biases in both dispersal and cooperation provides novel support for the predictions of the focal hypotheses, which are otherwise difficult to test given the limited variation in dispersal sex biases within taxonomic classes [3–5]. While our finding of female-biased natal cooperation is unusual for cooperative birds, this is principally the case because in many cooperative bird species only males delay dispersal and so only males are available to help within the natal group [2]. Among those cooperative bird species in which both sexes delay dispersal from their natal group and help, significantly female-biased rates of helping with one or more forms of care while within the natal group now appear to have been documented in a comparable number of species to significantly male-biased rates of helping [4]. Below, we consider the potential for the Philopatry and Dispersal trade-off hypotheses to account for the evolution of female-biased natal cooperation in this species, drawing on our empirical tests of the underlying mechanisms assumed by each. Our findings suggest that the mechanisms envisaged in the Philopatry hypothesis are unlikely to be acting in this species, and that the evolution of female-biased cooperation can be more readily explained by the Dispersal trade-off hypothesis. We consider the wider implications of this finding for our understanding of the evolution of sex differences in cooperation and, more broadly, the role that direct benefits of helping have played in the evolution of cooperation across taxa.

Among the diverse hypotheses proposed for the evolution of sex differences in cooperation [2,4], the Philopatry and Dispersal trade-off hypotheses are unusual in predicting the evolution of female-biased natal cooperation in sparrow-weaver societies; alternative hypotheses cannot readily explain this pattern. That the observed sex difference in natal cooperation occurs in the absence of a sex difference in average relatedness to recipients suggests that it cannot be attributed to a sex difference in the indirect benefits of cooperation [1,3,4]. While the indirect

benefits of cooperation are also a product of the effect of a given unit of cooperation (e.g., a given provisioned food item here) upon the fitness of the recipient (b in Hamilton's rule; [30]), it is not likely that sex differences exist in this; items of provisioned food are expected to impact the fitness of recipients regardless of the sex of donor. The heterogamety hypothesis [2,4,31] proposes that effects of the sex chromosomes on patterns of genetic relatedness could leave the heterogametic sex (females in birds) less cooperative, yet we see the opposite pattern here. Similarly, the paternity uncertainty hypothesis [2,4,32] predicts male-biased cooperation when extra-pair paternity occurs, yet we see the opposite pattern here despite dominant males losing 12% to 18% of paternity to other (i.e., extra-group) dominant males [20]. The parental skills hypothesis [2,4] proposes that the sex that invests most in parental care could contribute most to helping if (i) helping yields a direct benefit by improving parenting skills; and (ii) this benefit is larger for the sex that contributes most to parental care. This hypothesis could be relevant to sparrow-weavers as dominant females feed offspring at higher rates than dominant males [27]. However, compelling evidence that helping improves parenting skills remains elusive [2,33], and as immigrant subordinate sparrow-weavers rarely help at all [27], it seems unlikely that direct benefits arising via skills acquisition are a major driver of helping in this system. It has also been suggested that the sex that shows higher variance in lifetime reproductive success (LRS) may invest more in helping given its lower chance of securing direct fitness by breeding [2,34]. However, male and female sparrow-weavers likely have very similar variances in LRS, as both sexes only breed as dominants and do so in socially monogamous pairs [20] with no evident sex difference in dominance tenure length (t = −0.25, df = 62, $p$ = 0.800, $n$ = 32 male tenures, 32 female tenures, means ± SE: males = 2.62 ± 0.32 years, females = 2.51 ± 0.33 years). If anything, competition for extra-group paternity among dominant males [22] may leave males with slightly higher variance in LRS than females, but this would lead to the prediction of male-biased natal cooperation, and we observe the reverse. It is also unlikely that female helpers cooperatively provision more because male helpers contribute more to other cooperative activities (i.e., a sex-based division of cooperative labour) because female helpers contribute just as much as males to cooperative sentinelling and territory defence [25,35]. As helping may improve the survival of the dominant female [28] (by lightening her provisioning workload [29]), another possibility is that a sex difference exists in the direct cost to subordinates of helping to do so. For example, the survival of the dominant female is a direct barrier to the subordinate female inheriting the dominant breeding position and so subordinate females may pay a greater direct cost of helping via this mechanism (particularly as they may on average stay for longer in their natal groups than males; Fig 1A). However, this would lead to the expectation that subordinate females contribute less (rather than more) to cooperative care. It also is not clear that a sex difference in this cost exists, as subordinate males could face a similar direct cost of prolonging dominant female survival (as a son becoming a natal breeding dominant might also be contingent upon the death of their mother if she avoids close inbreeding [23]).

The Philopatry hypothesis proposes that offspring of the more philopatric sex contribute more to natal cooperation because, by staying in their natal group for longer, they stand to gain a greater downstream direct benefit from natal cooperation [1,3,6]. The commonly invoked mechanism by which the more philopatric sex could gain a greater direct benefit from cooperation is if they were more likely to ultimately breed in their natal group and thus benefit from having helped to increase the size of the natal workforce [1,3,6]. This mechanism cannot readily explain female-biased natal cooperation in sparrow weavers, as neither sex breeds within their natal group while subordinate [20], and our findings suggest that subordinate females are no more likely than males to inherit a dominant breeding position within their natal territory. Indeed, our findings also give cause to question whether helping in this species yields any form of direct benefit contingent upon inheriting natal dominance. If it did, one

might expect the markedly higher chance of inheriting natal dominance beyond 3 years old (Fig 2) to be accompanied by a concomitant increase in natal cooperation, yet we observe the opposite (Fig 1B). This pattern may instead reflect subordinates showing greater restraint from costly helping once they become credible contenders for dominance [36]. A second mechanism by which the more philopatric sex might gain a greater downstream direct benefit from natal helping, regardless of whether they inherit natal dominance, is if (i) helping to rear offspring augments natal group size; and (ii) living in a larger natal group then improves survival. However, this mechanism is also unlikely to apply in sparrow-weavers as (i) helping does not appear to augment group size (as helper numbers do not positively predict the mean rate of offspring survival or the breeding rate of dominant females [19]); and (ii) subordinate sparrow weavers appear to suffer markedly lower survival rates in larger social groups, suggesting that group augmentation would actually yield direct fitness costs rather than benefits [28]. Indeed, it seems unlikely that helping yields an appreciable direct benefit of any kind in sparrow weaver societies (e.g., via other mechanisms such as "pay to stay" or signalling "quality" [37–40]), as subordinates rarely contribute to cooperative care following dispersal from their family groups [20,27]. Together, our findings suggest that the Philopatry hypothesis cannot readily account for the evolution of female-biased natal cooperation in this species.

The Dispersal trade-off hypothesis proposes that the more dispersive sex contributes less to natal cooperation because all individuals face a trade-off between activities that promote dispersal (such as prospecting for dispersal opportunities) and natal cooperation [5,9]. It thus envisages that sex differences in cooperation arise not from a sex difference in the direct benefits of cooperation but from a trade-off between cooperation and other fitness-relevant traits in which sex differences exist. Our findings support the key prediction of this hypothesis, that male-biased dispersal should be accompanied by female-biased natal cooperation, and provide evidence in support of the underlying mechanisms that it envisages. First, our automated radio-tracking study revealed that natal subordinates of both sexes prospect, but males (the more dispersive sex) prospect at significantly higher rates than females, while making forays of comparable duration and distance. Our direct observations of prospectors visiting our study groups also reflect this male-bias in prospecting incidence. Subordinate males may benefit more from prospecting than females because they are more likely than females to disperse in all age classes, and because they disperse further on average than females between their natal and breeding sites [21]. While forays in some species function in extra-group mating [18,41,42], the forays of subordinate sparrow-weavers do not appear to serve this purpose as both within- and extra-group paternity are monopolised by dominant males [20,22]. That this male bias in prospecting was apparent during nestling provisioning periods highlights its potential to drive sex differences in cooperative care. That the sex difference in prospecting was also apparent outside nestling provisioning periods suggests that it is unlikely to be a product of the sex difference in natal cooperation (i.e., a reversal of the causal direction suggested above), as it also occurs when subordinates are not engaged in cooperative care.

Our automated radio-tracking study also revealed support for the second key assumption of the Dispersal trade-off hypothesis: that investments in prospecting trade-off against investments in cooperative care. Despite the modest sample size available, our analysis revealed that individual variation in prospecting rate significantly negatively predicts cooperative provisioning rate, consistent with the expectation of a trade-off between the two. Thus male and female sparrow weavers may differ in their mean contributions to cooperative care because they differ in their mean prospecting rate and the two traits are subject to a trade-off (the rationale of the Dispersal trade-off hypothesis). As prospecting and cooperative care are both likely to entail energetic costs [9,12,17] and are mutually exclusive activities (feeding within the natal territory cannot occur while prospecting elsewhere), the observed negative association between the two

traits could reflect a simple resource and/or time allocation trade-off (potentially mediated by endocrine changes associated with prospecting [9,42]). While forays could well entail intense energetic expenditures and will reduce the time available for feeding, the total energetic and/or time cost of prospecting in this species need not necessarily be high, as the detected forays occurred on average at modest daily rates (though our analyses could underestimate this) and were often short in duration. However, prospecting is also likely to entail a significant risk of injury and death, due to attacks by conspecifics [5,9,12,13] and predators [15]. As such, regular prospectors may benefit more than others from maintaining a competitive phenotype (to mitigate the risk of injury and enhance dispersal success), which could itself trade-off against investments in cooperation, compounding the effects of any energetic and/or time costs entailed in prospecting. As both of these trade-off mechanisms are consistent with the rationale of the Dispersal trade-off hypothesis, this hypothesis seems well placed to explain the evolution of female-biased natal cooperation in this species: males contribute less to natal cooperation because male-biased dispersal leaves them prospecting more, and all individuals face a trade-off between prospecting and natal cooperation.

Our study has tested the predictions and assumptions of the Philopatry and Dispersal trade-off hypotheses for the evolution of sex differences in cooperation. Our findings suggest that the mechanisms envisaged in the Dispersal trade-off hypothesis are better placed to explain the evolution of female-biased natal helping in sparrow-weaver societies than those envisaged in the Philopatry hypothesis. This is important because the Philopatry hypothesis is commonly invoked as the likely driver of the association between sex differences in Philopatry and cooperation across taxa (e.g., [1,3,4]), but the Dispersal trade-off hypothesis also predicts this pattern [4,5,9]. Moreover, the Dispersal trade-off hypothesis could apply more widely across taxa, as it does not require that helping yields a direct fitness benefit contingent upon remaining in the natal group, a scenario whose generality is unclear. The mechanisms envisaged in the Philopatry hypothesis could certainly apply in some species, such as those in which helping augments group size and in which group augmentation could yield a downstream direct benefit to helpers (e.g., Florida scrub jays, *Aphelocoma coerulescens* [6] and meerkats [1]). Indeed, the mechanisms envisaged in both hypotheses could act in concert to select for sex differences in helping in such species (e.g., both Florida scrub jays and meerkats also show sex differences in helper prospecting [6,9,18]). Our findings highlight the need for caution, however, when extrapolating the Philopatry hypothesis to species in which helping may not yield direct benefits or in which mechanisms for the differential downstream accrual of such benefits by the more philopatric sex have not been identified. On balance, it seems likely that the mechanisms envisaged in the Philopatry and Dispersal trade-off hypotheses have both played a role in the evolution of sex differences in natal cooperation, but their relative importance may vary markedly among species.

Our findings have wider implications too for the role that direct benefits of helping may have played in the evolution of cooperation. Recent comparative studies have highlighted that the probability that helpers will breed within their natal group predicts helper contributions to cooperative care across taxa [3,10], and it has been suggested that these findings constitute rare evidence that direct fitness benefits of helping have played a widespread role in the evolution of helping, alongside kin selection [3,4,10]. However, our findings suggest that such associations could also arise in the absence of direct benefits of helping, via trade-offs between dispersal and cooperation: helpers with higher probabilities of natal breeding might generally contribute more to natal cooperation across taxa because they need not invest as much in preparations for dispersal. Thus, while direct benefits of helping may indeed be important in some contexts (e.g., [1,39,40]), how widespread a role they have played in the evolution of helping behaviour across taxa would seem to remain an open question.

## Methods

### Study system and life-history monitoring

This work was conducted as part of the long-term field study of white-browed sparrow weavers in Tswalu Kalahari Reserve in the Northern Cape Province of South Africa (27˚16'S, 22˚25' E). Fieldwork was carried out from September to May between 2007 and 2016 inclusive. All social groups within a study area of approximately 1.5 km$^2$ were monitored each year; approximately 40 groups, each defending a small year-round territory. Throughout the study period, all birds in the study population were fitted with a single metal ring, 3 colour rings for identification (under SAFRING license 1444) and genotyped using 10 microsatellite loci (see Supplementary text A in [19,20] for genotyping protocols and relatedness calculations). The birds could be reliably caught from their woven roost chambers at night using custom-made sweep nets and were then returned to their chambers to pass the remainder of the night on completion of any processing. Social groups were easily distinguishable as all group members foraged together within their territory, defended it against neighbouring groups, and roosted within woven chambers in a single tree or cluster of trees close to the territory centre [20,35].

Each sparrow weaver group contains a single dominant (reproductive) pair and from 0 to 10 nonbreeding subordinates [20]. Dominants are easily distinguished from subordinates because they display a distinct set of behaviours [35], and the sexes are easily distinguished after the first 6 months of life because the subspecies that we study (*P. mahali mahali*) exhibits a beak colour sexual dimorphism [43]. Group compositions were assessed every week in the morning throughout each field season [19] and were confirmed by periodic captures of all group members from their roost chambers at night. We also sought to confirm the dominance status of birds on all such visits. Once a bird acquired a dominant position they retained dominance for life. Dominance tenures typically ended with the bird's immediate disappearance from the study population; a scenario that likely reflects their death, given the very local nature of dispersal in this species and the relative rarity with which unmarked adults enter our study population from outside [21]. On rare occasions dominants did transfer into a dominant position in another group, but on no occasion was a dominant bird known to subsequently become subordinate (either in their own group or any other). Dispersal was deemed to have occurred when an individual that was previously resident in one group (i.e., foraging with the group in the day and roosting on their territory at night) transferred to become resident in another group. All offspring delay dispersal from their natal group and those that do disperse typically make just 1 or 2 dispersal movements in their lifetimes [20,21]. From October 2011 onwards, we also noted any observations of visits to our focal groups by birds not resident on the focal territory (i.e., prospecting birds), determined via their colour ring combination and/ or our knowledge of the current locations of all resident birds. Visiting birds were typically subjected to aggressive chasing, which often hampered identification. Throughout each field season, we also checked every territory for nests (conspicuous woven structures with a different shape to roosting chambers) every 1 or 2 days, and wherever nests were present we checked them for the presence of newly laid eggs using an endoscope. When new eggs were found, we monitored egg laying through to clutch completion and the fate of the brood through to fledging [20]. We then monitored the rates at which all group members provisioned the growing nestlings via the use of video cameras placed below the focal nest between the 6th and 12th days after the first egg in the focal clutch had hatched. For full details on the monitoring and calculation of individual provisioning rates, see Text A in S1 File. Ethical approval for all protocols was provided by the University of Exeter and the University of Pretoria Animal Ethics Committee (EC023-07, EC100-12, EC007-17).

## Quantifying the prospecting behaviour of natal subordinates

We used Encounternet [44,45] to identify extra-territorial prospecting movements by natal subordinate sparrow weavers. This system is a wireless sensor network formed by an array of stationary radio receivers (termed "base stations") that log the radio-signals of any animal-mounted transmitters (termed "tags"; used here in transmit-only mode) within range. The logs collated by the base stations provide information on the tag (and therefore animal) ID detected, the date and time of detection and the signal strength (expressed as received signal strength indicator [RSSI] values). Between February and April 2017, we fitted Encounternet tags to 32 adult subordinate birds, using a figure-of-eight leg harness made of a stretchable, porous material that was stuck to the tag. The total mass of the tag and harness was approximately 1.2 g (2.6% to 3.1% of the body mass of the tagged birds at fitting; Fig 3A). The birds were tracked for an average of 37 days (range = 11–59 days) generating a total of 22,518,022 detection logs (individual base-stations log all tags within detection range every 5 s). Field observations after tag fitting confirmed that none of the tagged birds showed any signs of affected behaviour or impaired movement, and our results suggested that they still regularly engaged in extra-territorial movements. All tags were retrieved from the birds by recapture before the end of April 2017. The tagged birds were all (nonbreeding) subordinates that were resident on their natal territory (hereafter, their "home" territory) at the time of tagging and remained so for the duration of the Encounternet deployment. All tags were set to transmit ID-coded pulses every 5 s. To track the locations of the tags across our study site, we set up an array of 35 Encounternet base station receivers, each placed in the central tree (typically the main roost tree) of 35 contiguous sparrow weaver territories (S1 Fig). See Text B in S1 File for further detail on Encounternet deployment and base station calibration.

We processed Encounternet logs for each tag separately, working through the continuous time series of logs for each tag to estimate the locations of the tagged bird throughout the tag's deployment. A given tag's logs were processed in moving windows of 15 s and in time steps of 5 s. For each of these 15-s windows, we attempted to assign the tagged bird a "best estimate" location following a conservative set of rules depicted as a flow diagram in S3 Fig (see also Text B in S1 File). We then processed the resulting time series of "best estimate" locations for the tagged bird to identify extra-territorial prospecting forays using a conservative approach that should minimise the chance of false positives (S3 Fig and Text B in S1 File).

Using this data set, we identified prospecting forays as events that satisfied both of the following criteria: (i) a single "time period" or set of contiguous time periods for which the tagged bird had been assigned known "best estimate" locations that were not the tagged bird's home territory for a total of >15 s; and (ii) the furthest assigned "best estimate" location during this contiguous non-home period was >250 metres away from the centre of their home territory (we set this distance threshold to >250 m a priori, see Text B in S1 File for further details). This approach was designed to minimise the chance that the bird's territorial interactions with its neighbours while "at home" are misclassified as extra-territorial forays. While setting the distance threshold at 250 m makes sense a priori based on the biology of the bird (Text B in S1 File), we verified via a sensitivity analysis that our key finding of a sex difference in prospecting rate was not specific to this threshold setting (S4 Fig).

For each foray, we estimated "foray duration" as the time elapsed between the focal bird's first "best estimate" location outside their home territory and their subsequent first "best estimate" location back on their home territory. We estimated "foray distance" as the linear distance between the bird's home territory base station (approximately the centre of their home territory) and the furthest "best estimate" location that they were assigned during the foray. These "foray distances" could therefore underestimate the true distance travelled, because (i)

the bird could have travelled on a more complex path; and (ii) the furthest "best estimate" location could be closer to home than the true furthest location visited, as the former was constrained to be a location within our base station array. If prospectors did travel beyond the boundary of our array, their forays would still be detected (despite having potentially long periods of "unknown" locations while far from the array) if the bird passed within receiving range of one or more base stations >250 m from their "home" station during their foray. The final data set included extra-territorial forays for 27 tagged birds (Text B in S1 File).

## Evaluation of potential sampling biases

We used STRANGE framework for animal behaviour research [46] to evaluate if sampling biases could affect the generalisability of our findings. In short, cooperative provisioning effort of all subordinate individuals per social group was routinely monitored throughout the study period, without selecting which individuals were measured based on their size, social background, genetic make-up, or life experience [46]. Similarly, for Encounternet deployment, we tagged male and female subordinates residing across the entire field site of similar age per social group but not selecting individuals based on their prospecting behaviour, size, social background, genetic make-up, or life experience [46].

## Statistical methods

We employed a full model approach, evaluating the statistical importance of each predictor using likelihood-ratio tests (LRTs). This is a conservative approach with respect to statistical significance and provides model estimates for all variables of interest [47]. If not statistically significant, interactive terms were removed from initially specified full models to ease the interpretation of the effects of noninteractive terms. All noninteractive fixed effect predictors were retained within the full model regardless of their significance. All statistical analyses were carried out using R 4.3.1. [48]. Unless otherwise stated, statistical models were fitted using the R package "lme4" (v1.1.29; [49]). Over-dispersion and zero-inflation of generalised linear mixed models (GdLMMs) were checked by simulating scaled model residuals in the R package "DHARMa" (v0.4.6; [50]).

## Modelling the age-specific dispersal probabilities of natal subordinates

We investigated whether sparrow-weaver subordinates within their natal groups exhibit a sex difference in the age-specific probability of dispersal from their natal group, by fitting a binomial GdLMM to data reflecting whether or not they dispersed within a given year-long age window of their lives as a natal subordinate. All individuals hatched into our population between September 2007 and April 2014 that reached 6 months of age contributed to our analysis (individuals under 6 months of age very rarely disperse) and we utilised group census data up until April 2016 to determine if and when they dispersed (giving all individuals a minimum of 2 years of monitoring in which to disperse). The lives of each of the focal individuals were divided into year-long "age windows" from birth onwards. Each year-long age window that the individual started as a subordinate within its natal group (and before April 2016) contributed a line of data to the data set and was flagged to reflect whether they were (1) or were not (0) observed to have dispersed away from their natal group to another study group within our population during that year of life. Our analysis therefore focussed on known dispersal events (see Text C in S1 File for a discussion of unknown dispersal events). This approach resulted in a data set of 813 year-long windows for 338 individuals (178 males and 160 females) monitored as natal subordinates within 36 social groups. The full model contained the following fixed effect predictors: the subordinate's sex, age class during the focal year-long age window (as a

4-level categorical variable: <1 year old; 1 to 2 years old, 2 to 3 years old, and >3 years old; this last age class therefore captured any year-long age windows in which the focal individual was 3 years or older at the start of the age window) and the interaction between its sex and age class. Breeding season of hatching, individual ID, and natal group ID were included as random effect intercepts.

## Modelling the cooperative provisioning traits of natal subordinates

To investigate whether there is a sex difference in the nestling provisioning (helping) rates of natal subordinates, we fitted a zero-inflated negative binomial GdLMM using the R package "glmmTMB" (v1.1.7; [51]). We modelled provisioning rate (feeds/hour) by fitting the birds' "provisioning number" measure as the response term (the number of provisioning visits it conducted during a given approximately 3 h observation period; see above), with the duration of the observation period fitted as an offset. We included the following fixed effect predictors in the full model: the sex of the bird, their age (a 4-level factor: <1 year, 1 year to 2 years, 2 years to 3 years, and >3 years), the interaction between sex and age (given the possibility that the magnitude of sex differences changes with age), brood size (as a continuous variable, 1 to 3 nestlings), and brood age (a 7-level factor, 6- to 12-day old, given the potential for nonlinear effects). Breeding season, social group ID, clutch ID, and individual ID were included as random intercepts. The model was fitted to 1,338 observations of provisioning number by 314 different natal subordinates feeding 177 broods in 38 social groups. We found evidence for zero-inflation (zero-inflation parameter ± SE = −1.605 ± 0.181) and over-dispersion (dispersion parameter = 2.87), which were accounted for by our zero-inflated negative binomial model. To investigate whether there is a sex difference in natal subordinate provisioning visit durations (time spent within the nest chamber, from entry to exit, during which the provisioning bird interacts with the brood), we fitted a linear mixed model to the provisioning visit durations of the 5,034 provisioning events that natal subordinates conducted during the observation periods monitored above. The full model included the same fixed and random effect predictors as the analysis of provisioning rate described above. Provisioning visit duration was "ln+1" transformed to fulfil the assumption of normality in model residuals. To investigate whether there is a sex difference in the size of the food items that natal subordinates provisioning to broods, we fitted a binomial GdLMM to the binomial prey item size information (large or small) collected from the 1,325 provisioning events in which the focal bird's bill was visible prior to entering the nest (see above). We analysed the probability that the prey item was large, with a full model that included the same fixed and random effect predictors as the analysis of provisioning rates described above.

## Modelling the probability of acquiring dominance within the natal group

We investigated sex differences in the probability that natal subordinates inherit a dominant (breeding) position within their natal group using binomial GdLMMs. The data set comprised all birds that were known to be subordinate within their natal group at age 1, 2, 3, and 4 years, and we modelled whether or not they ultimately became dominant within their natal group. Subordinates were only included in the analysis if dominance turnovers were monitored within their natal group for a minimum of 2 years after the date on which the subordinate reached the focal age. Age, sex, and their interaction were included as a fixed effect predictor in these models. Breeding season of hatching and natal group ID were included as random intercepts. We also verified that similar patterns were reflected in a simple contingency table test for a sex difference in the proportion of all dominance acquisition events for a given sex in which the winner took dominance within their natal group. As subordinates could also

conceivably gain a downstream direct fitness benefit from natal helping if they take a dominance position via "territorial budding" from the natal territory (i.e., founding a new group on territory previously held by their natal group), we then repeated our modelling analysis above including within the "natal dominance inheritance events" events in which the focal bird did not inherit dominance within the natal group but instead became dominant by founding a new group via probable "territorial budding." Using the 88 dispersal events in the data set, we used binomial GdLMMs to explain the probability of dispersing into subordinate or dominant positions, including sex as a fixed effect predictor, and breeding season of hatching and natal group ID as random intercepts. We analysed dominance tenure length using a linear mixed model, including sex as a fixed effect and, breeding season of hatching and natal group ID as random intercepts. This model used data from 66 dominance tenures (32 males and 32 females) acquired by individuals hatched in the study population.

## Modelling the prospecting behaviour of natal subordinates

To investigate whether there is a sex difference in the daily rate of extra-territorial prospecting (number of forays per day), we employed a Poisson GdLMM. To investigate whether there is a sex difference in mean "foray duration" and mean "foray distance" (see above), we employed LMMs. In each of these 3 models, we included the same set of fixed effect predictors: age of the individual (as a continuous variable), sex, provisioning status of their home group (2-level factor [yes/no] to capture whether their home group was provisioning nestlings at the time), and the interaction between sex and provisioning status. In all 3 models, we included bird ID and home territory ID as random intercepts. In the model of the daily rate of prospecting, we also included observation ID as a random intercept, to account for overdispersion [52]. Sample sizes were as follows: prospecting rate analysis = 895 daily measures for 27 birds (13 male, 14 female) across 14 social groups; prospecting duration and distance analyses = estimates for 971 forays by 27 birds across 14 social groups.

To investigate the existence of a trade-off between cooperative nestling provisioning and prospecting, we tested whether these traits were negatively correlated at the individual level. For this analysis, we used the Encounternet data from the 13 tagged subordinates whose prospecting rates were quantified during a nestling feeding period within their group. For these birds we therefore had daily estimates of their prospecting rate, as well as estimates of their nestling provisioning rate from those mornings on which provisioning observations were also conducted (see above). To estimate a proxy for each bird's general level of prospecting effort, we calculated the bird's prospecting rate over a given number of days preceding each day of provisioning observations (see below). The full model included this proxy for the focal bird's prospecting effort as the main fixed effect predictor of interest in a GdLMM of daily provisioning rate (implemented using a Poisson process to model the number of nestling feeds conducted by a given bird during each observation watch and including the duration of the watch as an offset). We also included subordinate sex, brood age (as a continuous predictor in this case due to lower sample size, including this variable as a category does not affect the main conclusion of the analysis), and brood size as fixed effects, to ensure that independent effects of these variables were not confounding the prospecting effect of interest (given our use of a conservative full model approach, these terms were retained in the model regardless of their significance). Group ID and bird ID were included as random effect intercepts, given the repeated measures data structure. We did not find evidence for overdispersion in this model (DHARMa R package test: dispersion = 0.81, $p$ = 0.780).

The sample size for this trade-off analysis was modest (34 daily measures of provisioning rate for the 13 tagged subordinates with coupled prospecting rate estimates). However,

statistical power limitations are unlikely to have impacted our key inference from this analysis as the model nevertheless recovered the significant negative main effect of prospecting on provisioning rate that is expected under the Dispersal trade-off hypothesis ($\chi^2_1 = 5.08$, $p = 0.024$, Fig 3F and S9 Table). Given the very limited sample size available for sex comparisons within this data subset (6 males and 7 females), we did not seek to use this model to test for sex differences in cooperation (indeed, none would be expected under the Dispersal trade-off hypothesis while the effect of prospecting is being controlled). A sex effect was included in the model only to ensure that the prospecting effect detected was not confounded by uncontrolled sex differences in cooperation that arise for reasons other than prospecting. The analyses presented use the focal bird's prospecting rate calculated over a window of 3 days prior to the focal daily measure of their provisioning rate, as we thought it plausible a priori that any energetic or stress-related costs of prospecting might accumulate and be evident over this timescale. Recognising that the choice of 3 days is somewhat arbitrary, we confirmed through sensitivity analyses that the observed negative relationship between prospecting rate and provisioning rate was still evident when prospecting rate was calculated over a range of time window lengths (S2 Fig).

## Supporting information

**S1 File. Text A. Quantifying the contributions of natal subordinates to cooperative provisioning.** Text B. Additional Encounternet methods. Text C. Impact of unknown dispersal events on age-specific dispersal probabilities of natal subordinates. Text D. Sex difference in the strategy used to acquire dominance via dispersal.
(DOCX)

**S1 Fig. The Encounternet receiver "base station" array that was used for detecting forays.** Panel (**A**) and (**B**) show the same simplified map of the study site with the locations of the 35 base stations (black and red dots), most of which were placed in the centre of distinct sparrow weaver territories (see Methods). The x and y axes present longitude and latitude (respectively) in metres East and metres North of a given arbitrary location and thus also provide the scale for these maps. In each panel, a circle of 250 meters radius around a focal base station (red dot) is illustrated with a blue line; the only difference between the panels being the location of the focal base station. A single foray was defined as a continuous run (in time) of location estimates which suggested that the bird's closest base station was >250 m away from the centre of its home territory (i.e., outside blue circle) for at least 15 s. As the mean (± SE) distance between the centres of neighbouring territories was 93.7 m (± 4.56 m), the forays detected with this approach will typically have involved movements beyond the centres of the territories of neighbouring groups. This conservative approach will minimise the chance that a resident bird's territorial interactions with its neighbouring groups along their shared territory boundary are incorrectly interpreted as extra-territorial prospecting, but is likely to underestimate the true incidence of extra-territorial prospecting by excluding more local forays. The lack of base stations placed within the territories of study groups in the regions outside our core study population will also have left this approach underestimating true foray rate (and likely mean foray distance too). The land to the East and West of the presented array contains no other sparrow weaver territories within the pictured area (and so the focal birds will not have been conducting forays to groups living in those areas). However, there are a small number of widely spaced territories to the North of the array and several also lie close to the array to the South. Prospecting movements into these active territories will therefore not have been logged by our array. Spatial heterogeneity in the probability of true forays being detected by the array will have been accounted for in our statistical models of prospecting rate as all included social

Group ID (i.e., territory ID) as a random term. Data and code needed to generate this figure can be found at https://doi.org/10.5281/zenodo.13623047.
(PNG)

**S2 Fig. The effect of changing the number of days over which an individual's prospecting rate was calculated (x axis; prior to the day on which provisioning rate was measured) on the effect size estimate for the effect of an individual's prospecting rate (forays/day) on its cooperative provisioning rate (feeds/hour).** Our analyses within the main paper calculated the prospecting rate over the 3 days prior to the measurement of provisioning rate, as we thought it plausible a priori that any energetic or stress-related costs of prospecting might accumulate and be evident over this timescale. However, on recognising that this decision is somewhat arbitrary we sought to verify, via this sensitivity analysis, that the detected negative covariance between the 2 traits was not particular to this choice of time window. The analysis confirms evidence of negative covariance between the 2 traits over a range of time windows. The initial steady increase in the effect size as the length of the time window considered increases could reflect (i) the timescale over which accumulated costs arising from recent prospecting impact cooperative behaviour, and/or (ii) that, given the modest rate at which prospecting forays occur, the shortest time windows may simply give a poorer-quality estimate of the focal bird's overall true rate of prospecting. Dots and error bars represent mean model estimates ± SE for the effect of prospecting rate on provisioning rate from the model presented in S9 Table when calculating each bird's prospecting rate over different time windows. Data and code needed to generate this figure can be found at https://doi.org/10.5281/zenodo.13623047.
(PNG)

**S3 Fig. Decision tree applied to assign "best estimate" locations to tagged birds using the logs from the Encounternet receiver base station array (see Methods).** For each 15-s window in the time series of logs for a given bird, we attempted to assign the tagged bird a "best estimate" location via the following set of rules depicted by the flow diagram here. First, if there were no logs at all for the focal tag during the focal time window, we noted the bird's location as "unknown" (we did not draw spatial inferences from such "unknown" location events as they could reflect the bird being in a microenvironment that impeded signal transmission, such as thick cover). Second, if there were logs for the focal tag from just one base station (indicating that the tag was out of reception range from all other base stations), we assigned the tagged bird the location of the base station at which it was logged. Third, if there were logs for the focal tag from more than one base station but none of them were the tagged bird's home base station, we assigned the tagged bird the location of the base station whose logs had the highest mean signal strength. Note that the highest mean signal strength could nevertheless have been weak in this case (e.g., if the bird was well beyond the boundary of our base station array, having prospected out of the study area, its tag might be logged with only a weak signal strength at the base station closest to it on the array periphery; hence us terming these "best estimate" locations). Fourth, if there were logs for the focal tag from more than one base station but one of them was the bird's home base station, (i) if the home base station signal strength was stronger than −11.484 (the estimated signal strength at 50 m; see S5 Fig), we assigned the tagged bird the home base station location; (ii) if this was not the case, we assigned the tagged bird the location of the base station with the strongest mean signal strength.
(PNG)

**S4 Fig.** Sensitivity analysis to assess the effect of the "distance threshold" set during the foray detection process (see Methods) on (**A**) the total number of detected "forays" and (**B**) the effect

size estimate for the sex difference in prospecting rate (forays/day; males relative to females). (**A**) A "foray" was only considered to have occurred if the base station receiver that the focal bird was estimated to be closest to (i.e., its "best estimate" location) was further than a set distance threshold away from the base station at the centre of the bird's home territory (see Methods). A priori we set this distance threshold to be 250 m as this approach renders it highly likely that the focal bird itself is >125 m away from the centre of their home territory (see Methods in the main paper for the rationale). As the mean (± SE) distance between the centres of neighbouring territories is 93.7 m (± 4.56 m) in our study population, this minimum plausible distance of 125 m from the centre of the bird's home territory should ensure that "forays" detected using a 250 m distance threshold will typically have involved movements beyond the territory-centres of the tagged bird's neighbouring groups. This approach should thereby minimise the chance that the bird's territorial interactions with its neighbours along their shared boundary while "at home" are misclassified as extra-territorial forays. Reducing the distance threshold below 250 m will progressively increase the risk of such misclassifications; the likely cause of the marked increase in the number of "forays" detected when threshold distances of 225 m and 200 m are used (panel **A**), while increasing the distance threshold above 250 m may yield an excessively conservative approach that substantially underestimates the incidence of "true forays" by failing to capture those that occur over shorter distances. (**B**) The effect that changing this set distance threshold has on the estimated effect size (± SE) for the overall sex difference in prospecting rate (shown here as the effect of being male relative to female), when re-running the model presented in S6 Table using different distance thresholds during the foray classification process. The effect was consistently estimated to be positive across the range of distance thresholds tested (i.e., males having higher prospecting rates than females) and, as expected, the magnitude of the estimated sex difference effect size tended to increase as progressively higher distance thresholds were set. This is to be expected as lower distance thresholds will tend with greater frequency to misclassify home-territory movements (in which no sex difference is expected) as "forays," thereby obscuring to a greater degree our estimate of the sex difference in the true prospecting rate. All of the prospecting analyses presented and referred to in the main text used the 250 m distance threshold, as this makes most sense a priori from attention to the biology of the bird (see Methods). Data and code needed to generate this figure can be found at https://doi.org/10.5281/zenodo.13623047.
(PNG)

**S5 Fig.** The received signal strength indicator (RSSI) values (**A**) and percentage of detected signals (**B**) both decreased with the distance between tags and base-station receivers in a field validation on our study site. Our workflow for using the distribution of signal strengths across our receiver array to allocate "best estimate" locations for the tagged birds in each 15-s window, principally used information on the relative signal strength between receivers whenever tags were detected at multiple receivers simultaneously (see S3 Fig for details). However, to add an additional layer of conservatism to the assignment of "non-home" locations in scenarios in which a tag was detected by the receiver in the centre of the tagged bird's "home" territory as well as one or more receivers elsewhere, we also sought to estimate a threshold absolute signal strength that, if exceeded by the receiver on the home territory, would act as another indicator that the tagged bird was likely "home." To do this, we estimated the signal strength (RSSI)-distance relationship within our study site (panel A) using biologically realistic locations for the birds via the method described below, and then calculated the mean signal strength obtained at 50 m distance from a receiver in this context for use as this threshold value (as the mean ± SE distance between neighbouring territory centres is just 93.7 m ± 4.56 m in our study population). This process yielded a threshold RSSI value of −11.484; so if a

tagged bird was registered with an RSSI value above −11.484 at the receiver at the centre of its "home" territory, the bird was conservatively assigned a "home" location regardless of the signal strengths logged in other locations (see the final step in the S3 Fig workflow). We appreciate that the inherent variability within the RSSI-distance relationship (panel a; due to the effects for example of variation in tag height and signal obstruction via natural features on the study site) has 2 implications, and we do not consider either to be a problem. First, panel a highlights that tagged birds that are on their "home" territory will not always be registered at their "home" receiver at an RSSI >-11.484. In this scenario, we expect the other conservative aspects of our workflows for (i) determining the "best estimate" locations for birds (see S3 Fig) and (ii) identifying forays from the properties of any "non-home" locations (see main paper Methods) to ensure that the bird is not considered to be prospecting in this scenario. Second, panel A highlights that tagged birds could still be registered at their home receiver with an RSSI >-11.484 if they were 80 m from that receiver (potentially further if at heights in excess of those tested in this field exercise; see below) and thus potentially just inside the territory of a neighbouring group. We consider the assignment of a "home" location in this scenario appropriately conservative, as such locations could plausibly reflect routine territorial interactions between neighbouring groups rather than prospecting events. In order to characterise the relationship between RSSI and distance, we placed 2 tags on the end of a narrow wooden pole (one with its antenna oriented vertically, parallel to the base station antennae, and another with its antenna oriented horizontally, perpendicular to the base station antennae) and used the pole to move the tags among a series of locations on 10 transects, each starting from a different focal base station (itself hanging from a roost tree, just as the receivers in our array were). For each transect, we assessed the signal strength of the tags first at 2 m from the focal base station and then at locations at successive 10 m intervals up to a maximum of 122 m away from the focal base station. For one transect, the first location was accidentally set at 8 m from the focal base station and then sampled at 10 m intervals up to 108 m. At each location, we held the pole in position for 4 min, with tags located 1.70 m above the ground for 2 min, and then on the ground for 2 min, to simulate tagged birds perching in vegetation and foraging on the ground; the 2 activities and approximate heights that dominate their time budgets. Data and code needed to generate this figure can be found at https://doi.org/10.5281/zenodo.13623047. (PNG)

**S1 Table. Coefficients and likelihood-ratio tests of Gaussian mixed model explaining variation in the provisioning visit duration of subordinates within their natal groups (response variable, originally in seconds, ln+1 transformed; *n* = 5,040 provisioning visits by 205 subordinates, 97 males and 109 females).** The interaction between subordinate age and subordinate sex did not receive statistical support ($\chi^2_3 = 6.39$, $p = 0.094$) and was dropped from the full model to ease interpretation of single effect predictors. Residual variance = 0.262. (DOCX)

**S2 Table. Coefficients and likelihood-ratio tests of binomial mixed model explaining variation in probability of provisioning a large food item by subordinates within their natal groups (*n* = 1,325 provisioning visits by 156 subordinates, 74 males and 83 females).** The interaction between subordinate age and subordinate sex did not receive statistical support ($\chi^2_3 = 0.31$, $p = 0.859$) and was dropped from the full model to ease interpretation of single effect predictors. Model coefficients are shown in the link-function scale ("logit"). (DOCX)

**S3 Table. Microsatellite relatedness of subordinates and offspring (*n* = 487 relatedness measures for 205 subordinates, 111 males and 94 females).** This table shows the result of a

linear model explaining variation in microsatellite relatedness [5] between subordinates (males and females) and the offspring that they help to rear. In the model subordinate sex was included as a fixed effect predictor. For details of microsatellite genotyping, see Supplementary Text A in [6,7]. Residual variance = 0.041.
(DOCX)

**S4 Table. Coefficients and likelihood-ratio tests of binomial mixed model explaining variation in probability of subordinate dominance acquisition in the natal group when subordinates resided in the natal group at 1, 2, 3, and 4 years of age ($n$ = 375 age-specific observations, 114 males and 105 females).** The interaction between subordinate age and subordinate sex did not receive statistical support ($\chi^2_3 = 0.80$, $p = 0.850$) and was dropped from the full model to ease interpretation of single effect predictors. Model coefficients are shown in the link-function scale ("logit").
(DOCX)

**S5 Table. Coefficients and likelihood-ratio tests of binomial mixed model explaining variation in probability of subordinate dominance acquisition in the natal group when subordinates resided in the natal group at 1, 2, 3, and 4 years of age, including events any in which dominance was acquired outside the natal group by founding a new group within territory previously held by the natal group (i.e., territorial budding [8]) ($n$ = 375 age-specific observations, 114 males and 105 females).** The interaction between subordinate age and subordinate sex did not receive statistical support ($\chi^2_3 = 0.03$, $p = 0.870$) and was dropped from the full model to ease interpretation of single effect predictors. Model coefficients are shown in the link-function scale ("logit").
(DOCX)

**S6 Table. Coefficients and likelihood-ratio tests of Poisson mixed model explaining variation in prospecting rate (e.g., number of prospecting forays per day; $n$ = 895 daily measures of prospecting rate from 27 tagged birds).** No statistical support was found for the interaction between sex and provisioning ($\chi^2_1 = 0.97$, $p = 0.324$), which was removed from the final model. Model coefficients (Estimate) are shown along with standard errors (SE) and 95% confidence intervals (95% CIs). Model coefficients are shown in the link-function scale ("log").
(DOCX)

**S7 Table. Coefficients and likelihood-ratio tests of Gaussian mixed model (with log-transformed response variable) explaining variation in duration of individual forays (log minutes; $n$ = 971 prospecting forays from 27 tagged birds).** No statistical support was found for the interaction between sex and provisioning ($\chi^2_1 = 1.14$, $p = 0.285$), which was removed from the final model. Model coefficients (Estimate) are shown along with standard errors (SE) and 95% confidence intervals (95% CIs). Residual variance = 1.819.
(DOCX)

**S8 Table. Coefficients and likelihood-ratio tests of Gaussian mixed model (with log-transformed response variable) explaining variation in distance of individual forays (log meters; $n$ = 971 prospecting forays from 27 tagged birds).** No statistical support was found for the interaction between sex and provisioning ($\chi^2_1 = 0.142$, $p = 0.707$), which was removed from the final model. Model coefficients (Estimate) are shown along with standard errors (SE) and 95% confidence intervals (95% CIs). Residual variance = 0.059.
(DOCX)

**S9 Table. Poisson mixed model testing for a negative effect of prospecting rate (forays/ day) on cooperative provisioning rate (feeds/hour).** The data set comprised 34 daily

measures of provisioning rate (each with a paired estimate of prospecting effort) from 13 subordinates (6 males and 7 females) in 7 social groups. Prospecting rate was calculated in the 3 days leading up to the measurement of cooperative provisioning rate. Despite the modest sample size, the model recovered the significant negative main effect of prospecting rate on provisioning rate that is expected under the Dispersal trade-off hypothesis. The effects of subordinate sex, brood age, and brood size were included as fixed effects only to ensure that uncontrolled independent effects of these variables were unlikely to be confounding the prospecting effect of interest. Given our use of a conservative full model approach throughout, these terms were retained in the model regardless of their significance. With regard to the interpretation of the effect of subordinate sex, see footnote B. Model coefficients (Estimate) are shown along with standard errors (SEs) and 95% confidence intervals (95% CIs). (DOCX)

**S10 Table. Coefficients and likelihood-ratio tests of binomial mixed model explaining variation in the probability that natal subordinate individuals emigrate to a subordinate position (*n* = 88 dispersal events; 46 of 56 observed natal male dispersals; 19 of 32 observed natal female dispersals).** Model coefficients are shown in the link-function scale ("logit"). (DOCX)

## Acknowledgments

We thank the many team members who contributed to long-term data collection, Northern Cape Conservation for research permission, Nigel Bennett for invaluable logistical assistance, and E. Oppenheimer & Son, the Tswalu Foundation, and all at Tswalu Kalahari Reserve for their support in the field. We also thank Ben Hatchwell and Erik Postma for insightful discussions.

## Author Contributions

**Conceptualization:** Pablo Capilla-Lasheras, Andrew J. Young.

**Data curation:** Pablo Capilla-Lasheras, Nina Bircher, Antony M. Brown, Xavier Harrison, Andrew J. Young.

**Formal analysis:** Pablo Capilla-Lasheras.

**Funding acquisition:** Pablo Capilla-Lasheras, Andrew J. Young.

**Investigation:** Pablo Capilla-Lasheras, Nina Bircher, Antony M. Brown, Xavier Harrison, Thomas Reed, Jennifer E. York, Dominic L. Cram, Lindsay Walker, Marc Naguib, Andrew J. Young.

**Methodology:** Pablo Capilla-Lasheras, Nina Bircher, Antony M. Brown, Xavier Harrison, Christian Rutz, Marc Naguib, Andrew J. Young.

**Project administration:** Andrew J. Young.

**Resources:** Christian Rutz, Marc Naguib, Andrew J. Young.

**Supervision:** Xavier Harrison, Andrew J. Young.

**Visualization:** Pablo Capilla-Lasheras.

**Writing – original draft:** Pablo Capilla-Lasheras, Andrew J. Young.

**Writing – review & editing:** Pablo Capilla-Lasheras, Xavier Harrison, Thomas Reed, Jennifer E. York, Dominic L. Cram, Christian Rutz, Marc Naguib, Andrew J. Young.

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
