## [Editor Report · Decision Letter 0]

30 May 2024

Dear Dr Capilla-Lasheras, 

Thank you for submitting your manuscript entitled "Evolution of sex differences in cooperation: the role of trade-offs with dispersal" for consideration as a Research Article by PLOS Biology.

Your manuscript has now been evaluated by the PLOS Biology editorial staff, as well as by an academic editor with relevant expertise, and I'm writing to let you know that we would like to send your submission out for external peer review.

Once your full submission is complete, your paper will undergo a series of checks in preparation for peer review. After your manuscript has passed the checks it will be sent out for review. To provide the metadata for your submission, please Login to Editorial Manager (https://www.editorialmanager.com/pbiology) within two working days, i.e. by Jun 03 2024 11:59PM.

Kind regards,

Roli Roberts

Roland Roberts, PhD

Senior Editor

PLOS Biology

rroberts@plos.org

---

## [Decision Letter · Decision Letter 1]

1 Aug 2024

Dear Pablo,

Thank you for your patience while your manuscript " Evolution of sex differences in cooperation: the role of trade-offs with dispersal " went through peer-review at PLOS Biology. Your manuscript has now been evaluated by the PLOS Biology editors, an Academic Editor with relevant expertise, and by three independent reviewers.

You'll see that reviewer #1 is positive, while noting that the conclusions around the “dispersal trade-off” hypothesis are not iron-clad. S/he has a list of points that s/he would like you to discuss further. Reviewer #2 is extremely positive, and is impressed by the study, but has some presentational issues with the supplementary tables and asks about a potential error in a Figure. Reviewer #3 also enjoyed reading the paper, but raises a concern about the lack of tracking data from “non-prospector” individuals, and the impact this might have; s/he has a long list of further requests for clarification.

IMPORTANT: After some discussion, we think that your paper would be best considered as a Short Report. As your manuscript is already concise and has fewer than 4 Figures, no re-formatting is needed, but I've switched the article type in our system.

In light of the reviews, which you will find at the end of this email, we are pleased to offer you the opportunity to address the comments from the reviewers in a revision that we anticipate should not take you very long. We will then assess your revised manuscript and your response to the reviewers' comments with our Academic Editor aiming to avoid further rounds of peer-review, although might need to consult with the reviewers, depending on the nature of the revisions.

**IMPORTANT - SUBMITTING YOUR REVISION**

*Resubmission Checklist*

*Published Peer Review*

*PLOS Data Policy*

*Blot and Gel Data Policy*

Sincerely,

Roli

Roland Roberts, PhD

Senior Editor

PLOS Biology

rroberts@plos.org

REVIEWERS' COMMENTS:

Reviewer #1:

This is a nice paper arguing persuasively that the unusual pattern of female sex bias in provisioning behavior observed in the cooperatively breeding white-browed sparrow weaver is a consequence of a trade-off between the time subordinates spend looking for opportunities to disperse from their natal group and the time they have to feed young rather than that females are more likely to gain direct fitness advantages by inheriting dominance status in their natal group. The author's arguments are not iron-clad, but they're good and the extensive long-term data that they have obtained on this interesting species offers a strong argument supporting the "dispersal trade-off" hypothesis. In particular, their radio-tracking data, made possible by recent advances in tracking technology, provides novel a insight into this hypothesis. The authors do a good job of pointing out the potential wider implications of their results in that they counter the recent trend for claiming that direct fitness benefits play a widespread role in the evolution of helping behavior. 

Aspects of the authors' arguments that are worth additional discussion include:

1) The female bias in provisioning that they observe is indeed unusual, but largely because helpers in many species of cooperative breeders are primarily (if not exclusively) male. I don't know offhand whether this is also true for species in which both males and females serve as helpers relatively commonly; the authors should check. (They may have said this, but is the propensity for males and females to be subordinates the same; i.e., is the sex ratio of subordinates basically 50:50?)

2) The authors' argument regarding the greater effect of female subordinates is based on behavior, not fitness consequences. They do mention some aspects of the latter, but for the most part they say little about the role that subordinates play in enhancing reproductive success of groups or survivorship of dominants. The latter is potentially important in that if there is a difference it could result in significantly different future indirect fitness benefits of subordinate males compared to females. 

3) The differences in provisioning that they focus on also has potential relevance to the sex ratio via the repayment hypothesis (e.g. Emlen et al. 1986. Am. Nat. 127, 1-8). I don't encourage the authors to go down this rabbit hole, but it might be worth mentioning and is at least something the authors might want to look into at a later time.

4) I am concerned about the authors' claim that the dispersal trade-off hypothesis is supported by the sex difference in prospecting they observed even when birds are not provisioning nestlings. On the contrary, doesn't this suggest that some entirely different mechanism may be driving the difference in prospecting other than a trade-off with natal cooperation? 

5 I am aware of the difficulties of acquiring and dealing with the radio-tracking data that the authors are using to test the key prediction of the trade-off hypothesis. Despite the relatively modest sample size they were able to gather, their data is nonetheless valuable and important.

Specific comments:

Line 121. The female bias is, of course, not the case is a few avian taxa (such as waterfowl); perhaps this would be better stated as "a rare reversal of the typical passerine sex-bias in dispersal"

Line 129. I presume the authors are claiming that "female sparrow-weavers cooperatively feed offspring at substantially higher rates than subordinate males" primarily because female helpers are often rare or absent in cooperative breeders? If this assumption is correct, what about species where both males and females act as helpers? Is this statement still true? Are there more males subordinates that female subordinates?

Lines 131-134. Does feeding by subordinates increase reproductive success? Perhaps enhance survival rates by dominants? (In which case it would increase future indirect fitness.)

Lines 199-201. I realize that individual variation is not the focus of this paper, but were birds that fed more at the nest more likely to remain in their natal group and inherit dominance status?

Lines 202-205. What advantage to birds gain by dispersing to another group as a subordinate? Why not just stay in their natal group? 

Line 221. Are the differences still significant if individual birds (rather than forays) are considered independent points? (i.e., the bird was observed to foray or not). This seems like a fairer way to analyze these data.

Line 252. How is the test mentioned here ("whether the bird's social group was provisioning nestlings at the time" differ from the "nestling provisioning period" on line 250? The statistic cited here ("chi-square=0.97, p=0.32") does not match anything in Table S8.

Lines 258-264. These are indeed modest sample sizes, but I sympathize with the difficulty of acquiring such data and the authors have done a good job of qualifying what they have.

Line 304. It is not strictly true that a lack of any difference in average relatedness to recipients means that differences cannot be potentially attributed to indirect fitness effects; the authors have avoided discussing effects of helping on reproductive success of groups or (more cogently) survivorship of dominants. If helping results in increased survivorship of (related) dominants, females may be gaining future indirect fitness benefits à la Mumme et al. (1989, Helping behaviour, reproductive value, and the future component of indirect fitness. Animal Behaviour 38: 331-343.)

Lines 307-309. This implies that there is more than one dominant in a group? Or is there extra-group paternity, but only by dominants (as per lines 125-126)?

Lines 343-344. This appears to counter the argument (mentioned above) that helping might enhance breeder survivorship (or reproductive success?); if so, why do subordinates bother to provision at all (ever)? 

Lines 369-370. It's unclear to me why the authors seem to feel that the dispersal trade-off hypothesis is supported by the sex difference in prospecting being observed even when birds are not provisioning nestlings. To me, this suggests the possibility that some entirely different mechanism may be driving the difference in prospecting other than a trade-off with natal cooperation. 

Line 675. Delete "editors".

Fig. 2. I don't understand why the solid points are all way above where all the (obvious) translucent points are, especially for older subordinates.

Tables S12 and S13 are mirror images of each other; authors should pick one and delete the other. (Then again, it's a supplement, so maybe it doesn't really matter.)

Reviewer #2:

I am very pleased with this manuscript in every respect, and I would definitely rate it within the top 10% of manuscripts that I get to review. The research question is interesting and clearly explained, the empirical study is impressive and of the latest state of the art, the data analysis is remarkably careful and detailed, and the conclusions clearly follow from the data. The study provides strong support for the "Dispersal trade-off hypothesis" and thereby makes an important contribution to the research field. Moreover, the text is very well written and easy to understand. This leaves me with only a few relatively minor suggestions that I can make:

(1) For all mixed-effect models presented in the Supplementary Tables, please add a section of the table that covers all information about the random effects. I suggest to list each random effect (and the residual, where applicable), the size of the variance component, and the number of levels of the random factor. Finally, the legend should always give the number of rows in the analyzed data set. This would ensure that all relevant information can be found in one place (you could then consider shortening the details about number of levels of random effects in the Methods or Legends). This would also make it clear where an observation-level random effect (OLRE) was fitted to account for overdispersion (e.g. S8).

(2) For Table S8 it is stated that an OLRE was fitted to account for overdispersion in Poisson counts. In Table S11, you either may have forgotten to mention the use of an OLRE, or the count data was not overdispersed. Otherwise, I am a bit worried that the p-values in this Table might be incorrect (see Knief & Forstmeier 2021 in Behavior Research Methods).

(3) In Figure 1D, the estimates close to 0.55 seem to be far off from the data, since the density of raw data points at zero seems to be far greater than the density at 1. Therefore, the estimates should be far below 0.5. Also, the size of the SEs seems unrealistically large. Is this just an error during plotting, or is it that the plotting of model estimates does not work properly in a binomial mixed-model, especially when the random effects are highly influential? In the latter case, how about trying a Gaussian mixed model just for the purpose of plotting estimates that match the mean value of the data (see Knief & Forstmeier 2021)? If that advice turns out useful here, you may also want to double check other binomial models (e.g. Fig. 1A) for that issue.

Minor typos:

(1) Line 144: I think this should be "accompanied by female-biased natal cooperation"

(2) Y-axis of Fig. 1C: replace "log" with "ln". Likewise in Table S3 and in Line 569. Note that R uses the natural logarithm (ln) by default when using the "log" function.

(3) Line 476: "2.63.1%" contains a typo

Reviewer #3:

Overall I enjoyed reading this manuscript. There was some information left out of the introduction that was well covered in the discussion. Nevertheless, I do feel it should be added to the Introduction to increase clarity (see comments below). The study is interesting and overall robust. There are just a couple of issues of concern for me that may affect results and conclusions. Mainly: prospectors were tracked, these data seem good. However, conclusions were made about these movements without knowing how much non-prospectors move. Can we be sure that prospectors move more and have less time available to feed? This is assumed but no data given. Second, much is made of prospectors feeding young less, but the data appears biased here: provisioning rates were based on visits to the nest using nest observations overall, not when the individual was available to visit the nest. I have explained this in more detail below: but the males may in fact be feeding as much as the females at the times they are present. Not taking this into account will lead to a negative relationship between prospecting and feeding that may not be present. 

L 57-58: while it is true in some cooperative spp that one sex contribute more to care than the other, in other cooperative spp this is not the case, and should be acknowledged as such here to avoid this sounding like a general rule.

L 75: an additional potential explanation is that they may need to help in line with the pay-to-stay hypothesis

L 89" suggest change the word 'offspring' here - it assumes groups are simple family groups. In many species, they are complex groups of adults of varying relatedness that may decide to disperse or not, thus offspring is not always correct. 

L 107: unclear what is meant here by 'acting as required'

L 121: recent studies have found that in a number of avian species, the previously considered 'typical' sex biases in dispersal/philopatry do not occur. Instead, which sex disperses depends on sex ratios and opportunities in the larger population as a whole. See for example Speelman et al Anim Behav 208, 19-29, 2024, among others. 

L 129-131: where do you demonstrate this? There is no citation? Are you referring to the results of this paper, in the introduction? 

L 134-136: helps can also gain benefits of young surviving by greater predator detection in larger groups, defense of good quality territory etc that leads to greater survival - the focus here seems too narrow in terms of benefits

L 144: missing word. 

L 160: Main results should be in the main body text, not supplemental material: reader has to refer to SM to see the table output for this result. 

L 166: but is this relative to opportunity: did they feed more simply because they were there more often? Whereas on a per hour present basis, there was no difference? The figures suggest a prov rate per hour, but was this based on hours observing the nest, or hours that the helpers were present and able to provision young?

L 166-176: why is it called cooperative provisioning (which suggests it occured with another adult), and not just provisioning? Part way through it does change to just provisioning, and hence this is confusing. 

L 191: this raises the question of why there are sex differences in dispersal then, if females are no more likely to inherit? Can males gain EGP? 

L 221: but is your adult sex ratio 1:1? If not, then this is a moot point. 

L 235-237: but how different is this movement from typical (non-prospecting) movement in this species. Did you radio track non-prospectors to confirm they were not travelling similar distances (as in, territory boundaries are not stable in many species, and individuals can range widely throughout the day)?

L 239: 26 forays a day suggests this species may typically range widely, per above statement 

L 247-257: it is difficult to follow how these results were obtained, since a stats statement is given that refers to various models in SM, and some of the are the same table, some are different tables. For example, in this section, the results reported come from Tables S8, S8, S8, S9,S10,S9,S8,S10. This is very confusing jumps between model tables.

L 263-264: it's hard to understand what is meant here. A small sample size that was then varied between potential time windows based on what biological assumptions about appropriate time periods?

L 294: spelling - underling

L 309: are these extra group males? Or intragroup? Further down in methods this is explained, but at this point in the manuscript, the reader does not know, so would be good to clarify to avoid ambiguity.

L 375: see note above: this depends on if provisioning rate was calculated as number of hours nest observed, or number of hours individuals was present to provision the nest. If the latter not used, this biases the data

L 527: what is a GdLMM? I am used to acronyms GLMM and GLM but not GdLMM

L 522: the full model approach can lead to ambiguity and retention of many terms that are not significant predictors of data, as well as the predictor:sample size ratio leading to uncertain conclusions of significance. Justify use of the this approach relative to these concerns. 

Per comments above, the SM suggests that the provisioning rates were calculated from observations of the nest, in terms of next observation hours, not hours that the helpers was present (not prospecting) and available to help. This thus creates an issue: if a male is away 50% of the nest observation time, he will have a lower rate of provisioning per hour, even if he is not helping less. As in, in the hours he is there, he may be helping just as much. This could affect some of the conclusions made about this data. 

Tale S2: why was brood age categorical, not continuous? Brood age categorical, but brood size not? In Table S11, brood age seems continuous, whereas in other models it was not.

---

## [Editor Report · Decision Letter 2]

23 Sep 2024

Dear Pablo,

Thank you for the submission of your revised Short Reports "Evolution of sex differences in cooperation may be driven by trade-offs with dispersal" for publication in PLOS Biology. On behalf of my colleagues and the Academic Editor, Gail Patricelli, I'm pleased to say that we can in principle accept your manuscript for publication, provided you address any remaining formatting and reporting issues. These will be detailed in an email you should receive within 2-3 business days from our colleagues in the journal operations team; no action is required from you until then. Please note that we will not be able to formally accept your manuscript and schedule it for publication until you have completed any requested changes.

IMPORTANT: Before accepting your paper for publication, I made two changes to your manuscript file. They are:

a) Because we like to include active verbs in our Titles, and avoid punctuation, we've changed the Title slightly to "Evolution of sex differences in cooperation may be driven by trade-offs with dispersal." Do let me know if there's any problem with this. Note that because your supplementary file was a PDF, I was not able to change its title to match, so you will need to do this yourself.

b) Many thanks for providing the data and code in Zenodo. We need this to be cited in each relevant Figure legend, so I have included a statement directing readers to your Zenodo DOI in all relevant legends (i.e. all except Fig S3).

Sincerely, 

Roli

Senior Editor

PLOS Biology

rroberts@plos.org